# The Review of Chosen Methods Used to Investigate Heat Transfer in a Steel Porous Charge

Rafał Wyczółkowski [1,*], Vazgen Bagdasaryan [2], Marek Gała [3] and Paweł Artur Król [4]

1 Department of Production Management, Czestochowa University of Technology, Armii Krajowej 19, 42-200 Czestochowa, Poland

2 Institute of Civil Engineering, Warsaw University of Life Sciences—SGGW, Nowoursynowska 166, 02-787 Warsaw, Poland; vazgen_bagdasaryan@sggw.edu.pl

3 Institute of Electric Power Engineering, Czestochowa University of Technology, Armii Krajowej 17, 42-200 Czestochowa, Poland; marek.gala@pcz.pl

4 Institute of Building Engineering, Warsaw University of Technology, Pl. Politechniki 1, 00-661 Warsaw, Poland; pawel.krol@pw.edu.pl

* Correspondence: rafal.wyczolkowski@pcz.pl

**Abstract:** The paper presents chosen experimental and model methods of investigating heat transfer in a steel porous charge. The results of this investigation provide information on both the qualitative and quantitative course of the analysed processes of heat exchange. The parameters which characterise the analysed phenomenon in a quantitative manner, among others, are: The effective thermal conductivity $k_{ef}$, the thermal contact resistance $R_{ct}$ and Nusselt number $Nu$. It has been established that it is not possible to use literature models in order to determine the $k_{ef}$ coefficient. The authors present their own model of effective thermal conductivity. The above-mentioned parameters for a porous charge reach the values within the following ranges: $k_{ef}$: 1.0–8.5 W/(m·K); $R_{ct}$: 0.0019–0.0057 (m²·K)/W; $Nu$: 1.2–7.1.

**Keywords:** steel porous charge; effective thermal conductivity; Schlieren method; thermovision measurements; electric analogy

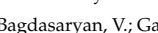



## 1. Introduction

Despite the dynamic development of materials science, steel remains the most important construction material, which is evidenced by its constantly growing global production [1]. The widespread use of steel as a construction material is due to its diverse mechanical properties, ranging from moderate 200–300 MPa yield strength with excellent ductility to yield stress of over 1400 MPa and fracture toughness of up to 100 MPa [2].

The main challenge for steel producers is to provide top-quality products obtained at the lowest possible costs. Heat treatment operations are the processes that essentially determine the quality of products supplied by the steel industry. Any shortcomings and errors in the heat treatment have a negative impact on further stages of the production, as well as the utility properties of the finished products. Therefore, the heat treatment processes need to be optimised, which is performed with the use of computers by means of model calculations [3–7]. One of the conditions for the uniqueness of such models is the thermophysical properties of the heated elements. In the case of solid elements, the basic thermal property of the treated charge is the thermal conductivity of steel $k_s$ [8,9]. However, in many situations, the treated charge has a porous structure [10,11]. Such charges are two-phase granular media with a solid metal skeleton and voids filled with gas. The type of this gas depends on the atmosphere of the furnace in which the heat treatment is carried out. Generally, we can distinguish three types of porous charge: bundles, coils and a piece charge. Bundles are used to treat various long products. This kind of charge can have an external or mixed porosity due to the geometry of individual elements. The bundles with

an external porosity relate to various types of bars (round, square or flat ones), whereas the charge, which is characterised by a mixed porosity, is made up of hollow components, such as pipes or rectangular sections [12,13]. The second group is coils, which are formed by heated sheets, tapes, wires or blanks [14,15]. Additionally, the third group, also called a piece charge, represents small products such as bolts, nuts, pins, pivots and many others, which are heated at the same time in large quantities in baskets or special containers [16,17]. Some examples of porous charges are presented in Figure 1.

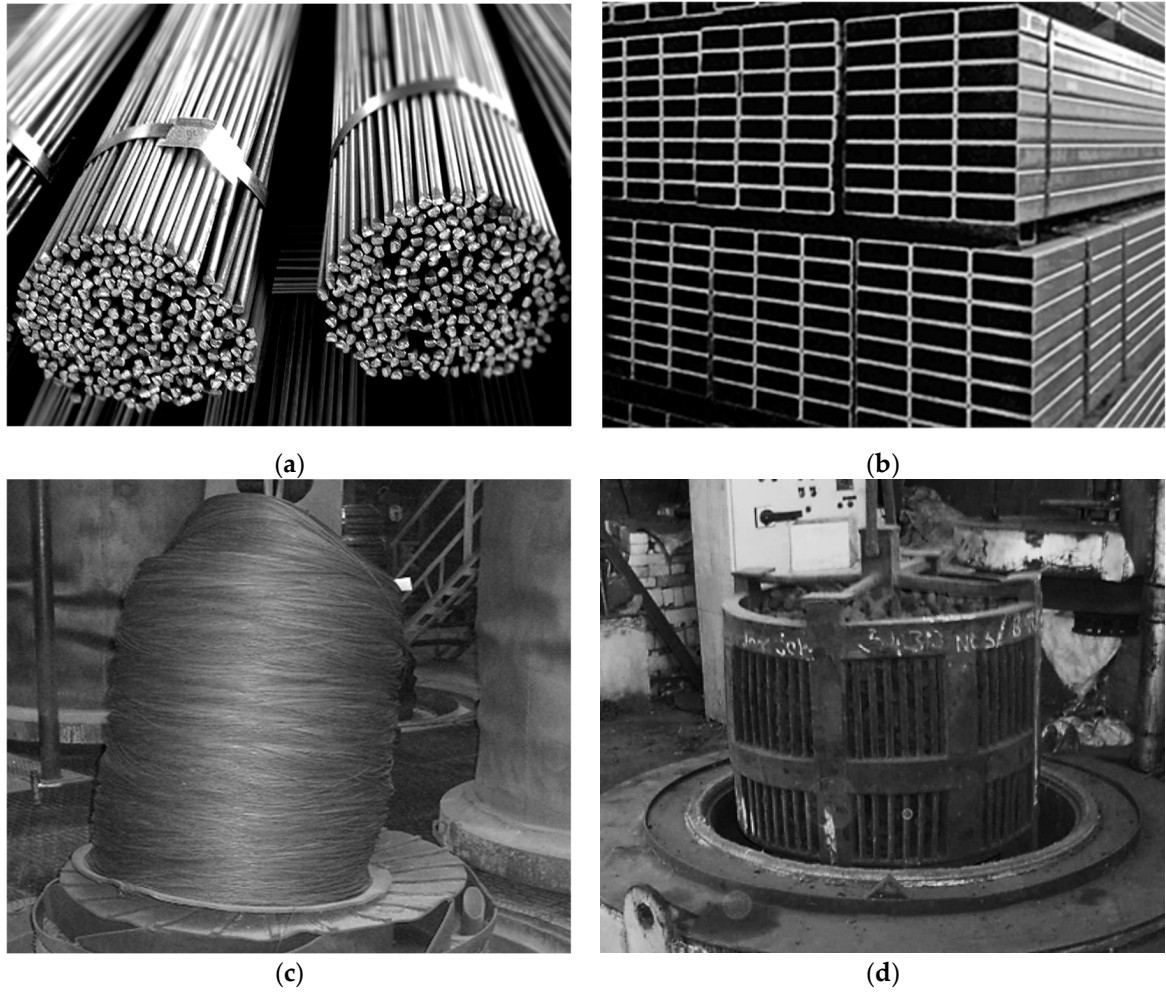

(a)          (b)

(c)          (d)

**Figure 1.** A steel porous charge in the form of a: (**a**) round bar bundle; (**b**) rectangular section bundle; (**c**) wire coil; (**d**) piece charge.

The basic qualities of a porous charge are: a granular structure resulting from the non-continuity of the solid phase and the presence of voids filled with gas. These features make the heating of a porous charge completely different from that of solid elements. In solid elements, heat transfer takes place by means of simple conduction, while in porous charges, it is determined by complex conduction phenomena in individual phases, contact conduction, thermal radiation and gas convection [18,19]. Therefore, the basic thermal property of a porous charge is an effective thermal conductivity $k_{ef}$. This parameter is widely used in the analysis of porous and nonhomogeneous media [20–22], and in contrast to the thermal conductivity of homogenous substances, it is not a material characteristic [23,24]. The value of the $k_{ef}$ coefficient of some granular media is a function of complex heat transfer mechanisms connected with their structure, chemical composition, and above all, with the temperature.

It is possible to determine the effective thermal conductivity by two methods: by means of experimental measurements and model calculations [25]. This paper presents a review of investigation methods used for determining the effective thermal conductivity of a steel porous charge. In the beginning, it discusses experimental investigations, the results of which provide quantitative information. After that, it deals with experimental methods which analyse the heat transfer in a porous charge from the qualitative point of view. The results of such investigations provide information which is necessary for mathematical modelling of effective thermal conductivity. The further part of the paper discusses the problem of model calculations of the effective thermal conductivity of a porous charge.

## 2. Measurements of the Effective Thermal Conductivity

Measurements represent the most reliable source of information about the value of the effective thermal conductivity of a porous charge. The $k_{ef}$ coefficient is determined with the use of standard methods applied for thermal conductivity measurements. In general, there are two basic methods for measuring thermal conductivity: steady-state methods and transient methods [26,27]. The steady-state methods record measurements when a tested sample reaches thermal equilibrium. This condition is attained when the temperature at each point of the tested sample does not change in time. This is the primary and most accurate method of thermal conductivity measurement. The main disadvantage of this technique is the long duration of measurements connected with reaching the required thermal equilibrium of the sample. The transient methods record a measurement during the heating process when the tested sample is in a transient thermal state. This means that the measurements can be performed relatively quickly, which is a major advantage in comparison with the steady-state techniques [28]. Taking into consideration the way of the heat flux flow through the sample, we can distinguish: (i) longitudinal methods that determine the temperature gradient in a sample with a uniform section, heated by one surface and cooled by the opposite one; (ii) radial methods—in which the temperature gradient is measured in the samples which surround entirely the source of heat. There are three main radial methods: cylindrical, spherical and elliptical. Due to the required temperature of the tests, measurement systems can be divided into three categories: (i) the ones operating at room temperature (20–25 °C), (ii) the ones operating below the room temperature (down to about −180 °C), and (iii) the ones operating at high temperature (up to 600 °C or above) [29].

For the measurements of the effective thermal conductivity of a porous charge, two steady-state methods are mainly applied: cylindrical and longitudinal ones, which apply Fourier's law of heat conduction. The measurements in the first case were performed with a hot pipe apparatus (HPA), in the second case with the use of a guarded hot plate apparatus (GHPA). There are some commercial apparatuses for determining the thermal conductivity based on the test methods—ISO, DIN, ASTM [30]. However, due to the peculiarity of the tested samples, original test stands built especially for this purpose are used for measuring the effective thermal conductivity of a porous charge.

### 2.1. Measurements with the Use of a Hot Pipe Apparatus

A hot pipe apparatus (pipe method) was used to examine the effective thermal conductivity of coils of a steel wire. The pipe method employs a radial heat flow in a cylindrical sample. A core heater (in the form of a tube, rod, or wire) is placed along the central axis of a pipe-shaped sample [28]. In this case, the tested samples are wire coils. These samples are obtained by winding the wire on a steel pipe, inside of which there is an electric resistance heater. The view of a fragment of such a sample made of a wire with a diameter of 0.9 mm is shown in Figure 2.

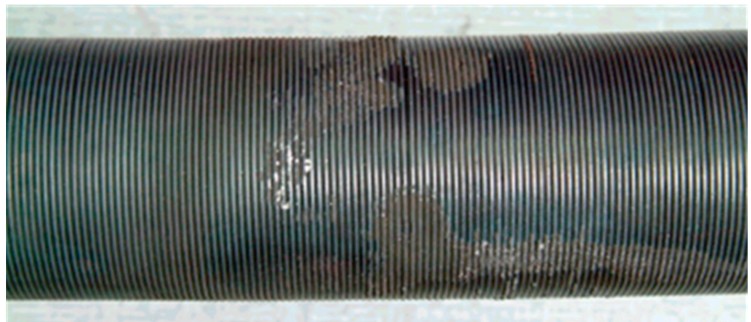

**Figure 2.** A view of the coil sample of a wire the diameter of 0.9 mm.

A custom experimental stand, designed as recommended by the ASTM standards [31], was used for the measurements. The details concerning the structure of the stand and the measurement procedure were described in [32]. The effective thermal conductivity determined with the use of this method is obtained from Fourier's heat conduction equation in a long cylinder by measuring the radial heat flux $Q$ in the sample [28]:

$$k_{ef} = \frac{Q \ln(d_2/d_1)}{2\pi(t_1 - t_2)},$$ (1)

where: $t_1$, $t_2$—temperature of the inner (hot) and outer (cold) surface of the sample; $d_1$, $d_2$—inner and outer diameter of the sample. Heat flux $Q$ flowing through the sample is determined based on the measurement of electric power consumed by the heater. This is possible because in the case of electric resistance heaters, the whole electrical energy is transformed into heat [33].

The measurements were performed for samples made of low-carbon steel wire with diameters of: 0.9, 1.5 and 2.0 mm for the temperature range of 50–700 °C. The maximum measurement uncertainty was 7%. The samples were wound with maximal coil density; therefore, their porosity was 0.2. The used wire was made from P2-04B blank with the chemical composition described by the certificate [34].

### 2.2. Measurements with the Use of a Guarded Hot Plate Apparatus

A guarded hot plate apparatus (GHP) was used to examine the effective thermal conductivity of a porous charge created from steel long elements such as: round or square bars and rectangular sections. GHP relies on a steady temperature difference over a sample with a known thickness, and its main aim is to control the heat flux $Q$ flowing through the tested material. For this reason, the samples of a porous charge tested by this method had the form of flat beds made up of specific types of long steel elements. Figure 3 shows the pictorial photos of the two samples composed of square bars and rectangular sections. Sample (a) represents a charge with an external porosity, whereas sample (b) represents a charge with a mixed porosity.

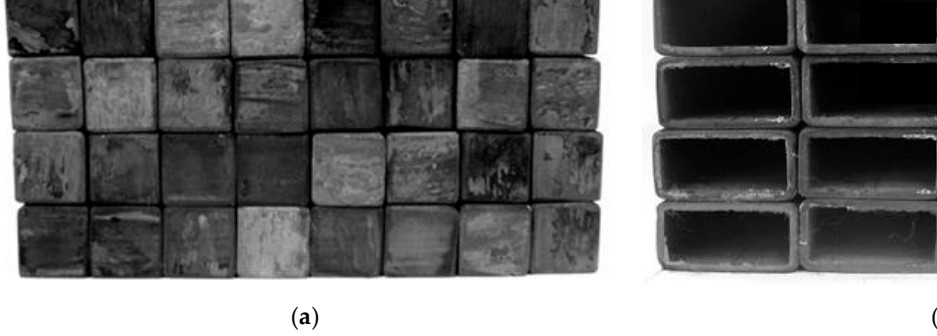

(**a**)　　　　　　　　　　　　　　　　　　(**b**)

**Figure 3.** Samples tested in the guarded hot plate apparatus: (**a**) square bars; (**b**) rectangular sections.

The measurements were performed with the use of a custom experimental stand, designed as recommended by the ASTM standards [35,36]. The details on the structure of the stand and measurement procedure, in this case, were described in [37]. The maximum measurement uncertainty in this stand was 4.6%.

The effective thermal conductivity determined with the use of this method is obtained from the Fourier's heat conduction equation in a plane wall [38]:

$$k_{ef} = \frac{QL}{\Delta t},$$ (2)

where: $Q$—heat flux flowing through the sample; $L$—sample dimension in the direction of the heat flow; $\Delta t$—temperature difference in the sample along the dimension $L$.

The samples of a porous charge investigated in GHPA were built with the use of the following components: round and square bars (10, 20, 30 mm) and rectangular sections (20 × 40, 40 × 40, 60 × 60 mm). All these components were made of S235JRH steel grade, which is characterised by a carbon content of 0.2% [39]. The test for the samples of round bars was performed for two arrangements: a staggered and an in-line one, as illustrated in Figure 4, whereas samples composed of square bars and sections had only an in-line arrangement as in Figure 3.

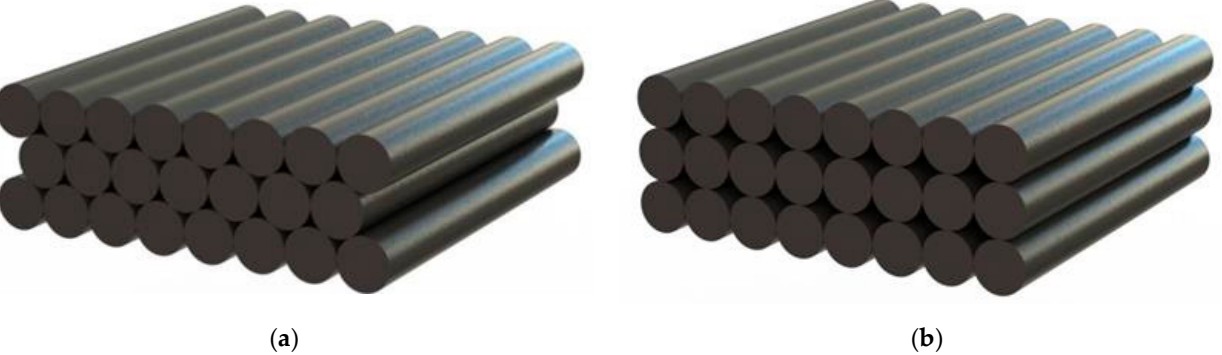

(**a**)　　　　　　　　　　　　　　　(**b**)

**Figure 4.** A staggered (**a**) and in-line (**b**) arrangement of the samples.

*2.3. Results of Measurements*

The results of the effective thermal conductivity measurements are shown in diagrams in the function of the mean measurement temperature $t_m$:

$$t_m = 0.5(t_{hot} + t_{cold}),$$ (3)

where: $t_{hot}$ and $t_{cold}$ are the mean temperatures of the hot and cold surfaces of the sample, respectively.

The values of the $k_{ef}$ coefficient obtained for the wire coil are shown in Figure 5. It can be seen that the effective thermal conductivity of this charge increases both with temperature and wire diameter. The values of the coefficient $k_{ef}$ for individual samples in the considered temperature range are: (i) 1.05–1.78 W/(m·K) for 0.9 mm wire; (ii) 1.61–2.57 W/(m·K) for 1.6 mm wire; (iii) 1.94–2.84 W/(m·K) for 2.0 mm wire.

The following figures show the results obtained for the measurements in the guarded hot plate apparatus. Figures 6 and 7 relate to the samples of round bars with a staggered and in-line arrangement, respectively. The values of the $k_{ef}$ coefficient in this case range 1.75–6.27 W/(m·K) (staggered samples) and 1.64–6.03 W/(m·K) (in-line samples). These results show the same tendencies of $k_{ef}$ coefficient changes as those obtained in the case of wire samples. The effective thermal conductivity grows together with temperature and bar diameter.

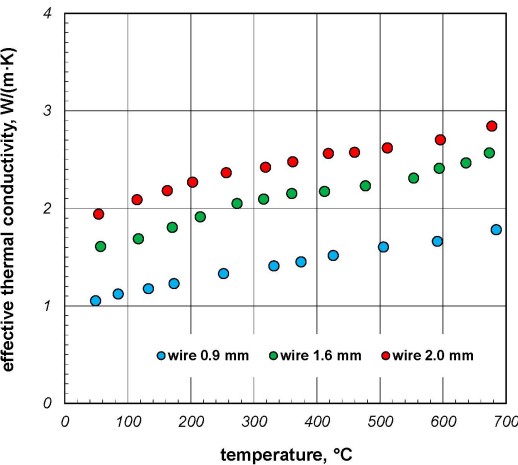

**Figure 5.** The effective thermal conductivity of the wire coil samples depending on the temperature and wire diameter.

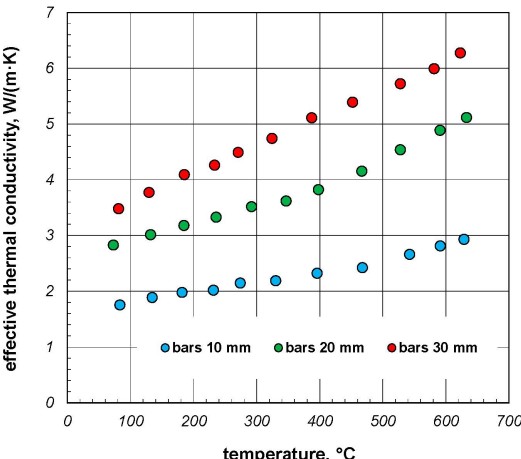

**Figure 6.** The effective thermal conductivity of staggered round bar samples depending on the temperature and bar diameter.

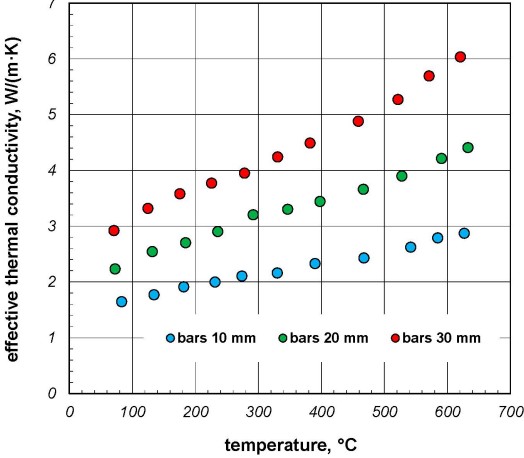

**Figure 7.** The effective thermal conductivity of in-line round bar samples depending on the temperature and bar diameter.

Figure 8 shows the results for the square bar samples. In this case, the $k_{ef}$ coefficient ranges from 2.69 to 8.19 W/(m·K). Compared to round bar samples, the $k_{ef}$ values obtained in this case are about 40% bigger. This increase results from the geometry of the charge;

when the bar surfaces are flat, the contact area for the adjacent bars is greater than in the bundle of round bars, which reduces the thermal contact resistance in the charge.

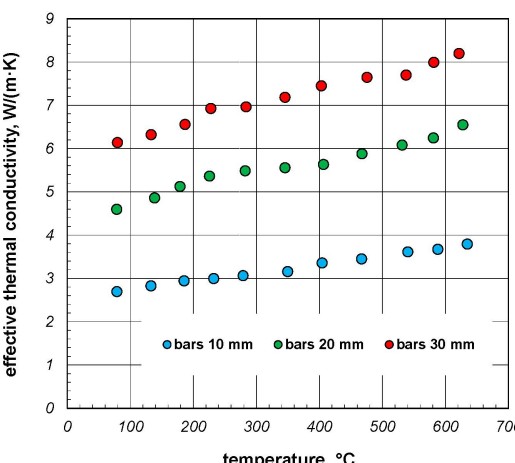

**Figure 8.** The effective thermal conductivity of square bar samples depending on the temperature and bar size.

Figure 9 presents the results obtained for section beds. The values observed for these samples were in the range of 3.63–6.31 W/(m·K), which is comparable to the previous samples. What is interesting, the value of the $k_{ef}$ coefficient did not grow with temperature for the sample of $20 \times 40$ mm sections. It ranged 3.63–3.85 W/(m·K), and its mean value was 3.73 W/(m·K). For the remaining samples of $40 \times 40$ mm and $60 \times 60$ mm sections, the $k_{ef}$ coefficient increased with temperature as it was in the cases of wire and bar samples.

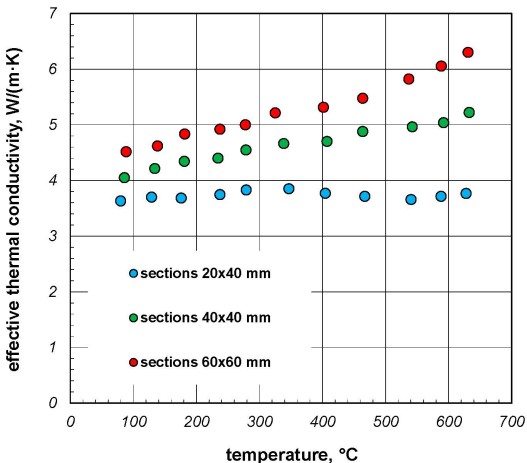

**Figure 9.** The effective thermal conductivity of rectangular section samples depending on the temperature and section size.

A comparative analysis of the results presented in Figures 5–9 shows that the effective thermal conductivity decreases with an increase in the charge porosity. Porosity's influence on the effective thermal conductivity was described in [37]. Moreover, the diagrams indicate that apart from one case, the effective thermal conductivity of the samples grows together with temperature. This is an opposite tendency compared to the steel thermal conductivity $k_s$, as this parameter falls together with temperature. The increase in $k_{ef}$ together with temperature is caused by the influence of heat radiation in cavities of the porous charge. Since the observed increase is linear, the results for individual samples were approximated by the method of the least squares with the use of a linear function:

$$k_{ef} = k_0 + \beta t, \tag{4}$$

The values of coefficients $k_0$ and $\beta$, as well as the coefficient of determination $R^2$, obtained for individual samples are presented in Table 1.

**Table 1.** The values of $k_0$, $\beta$ and $R^2$ coefficients for the investigated samples.

| Sample Type | Element Dimension | $k_0$ | $\beta$ | $R^2$ |
|---|---|---|---|---|
| Wire coil | 0.9 mm | 1.03 | 0.0011 | 0.995 |
| | 1.6 mm | 1.58 | 0.0013 | 0.974 |
| | 2.0 mm | 1.96 | 0.0014 | 0.971 |
| Staggered round bars | 10 mm | 1.57 | 0.0021 | 0.981 |
| | 20 mm | 2.39 | 0.0042 | 0.974 |
| | 30 mm | 3.11 | 0.0051 | 0.998 |
| In-line round bars | 10 mm | 1.48 | 0.0022 | 0.993 |
| | 20 mm | 2.02 | 0.0037 | 0.993 |
| | 30 mm | 2.54 | 0.0054 | 0.990 |
| Square bars | 10 mm | 2.54 | 0.0019 | 0.994 |
| | 20 mm | 4.48 | 0.0031 | 0.967 |
| | 30 mm | 5.91 | 0.0036 | 0.982 |
| Sections | 40 × 40 mm | 3.95 | 0.0019 | 0.984 |
| | 60 × 60 mm | 4.18 | 0.0031 | 0.977 |

The results from Table 1 indicate that the values of $k_0$ and $\beta$ coefficients were in the ranges of 1.03–4.18 and 0.0011–0.0054, respectively. What is more, the coefficient of determination $R^2$ is in the range of 0.967–0.998. The values of this parameter close to 1 show that the adopted equations are well fitted to the experimental results. The values of the $k_0$ coefficient clearly support the previous observations that the effective thermal conductivity of a specific type of porous charge grows with the increase in dimensions of the components. In the case of coils, this applies to the wire diameter to the bar diameter for bundles and to the transverse dimensions for square or rectangular elements. What is more, the values of the $\beta$ coefficient show that when the dimensions of the components increase, the temperature effect on the $k_{ef}$ coefficient is also stronger.

## 3. The Investigation of a Porous Charge by Means of Schlieren Imaging

Techniques related to optical visualisation are considered to be basic experimental methodologies used in examinations in the field of fluid mechanics and heat exchange. These methods utilise the effect of changes in the density of the examined medium on the refractive index, which makes them useful in examinations of transparent and non-luminous mediums [40,41]. This is a great advantage of the methods since they do not require adding special markers to the fluid to make visual observation possible. Furthermore, they also allow the researcher to examine many characteristics of the flow field available for visual perception, offering an insight into the physics of the phenomenon.

We can distinguish three basic methods of optical visualisation: shadow method, Schlieren (streak) imaging and interference method [42]. A very important role is played by the Schlieren method, which makes the gradient field image for refractive index visible and easy to be interpreted. The method is more sensitive than the shadow method and easier for qualitative visualisation compared to the interference method. In physical terms, the Schlieren method is based on the angular deviation of light rays that go through the area of the transparent fluid that has an inhomogeneous refractive index $n$. Gradients of the coefficient $n$ are caused by the inhomogeneity of temperature, density and concentration of various components. Convection examinations use the effect of inhomogeneity of density of the fluid caused by the varied temperature field on light distribution [42]. The devices that use this phenomenon for visualisation of the inhomogeneity of the medium are called

Schlieren apparatuses. Changes in light intensity are proportional to the first derivative of fluid density [43]:

$$\frac{\Delta I}{I} \approx \frac{d\rho}{dx},$$

(5)

where: $\Delta I/I$—relative light intensity after passing through the optical system of the Schlieren apparatus; $\rho$—density of the medium; $x$—direction of the normal to the surface with a similar material density.

The use of the Schlieren imaging for examinations of a porous charge was presented for the samples made of rectangular sections. Since this type of charge is characterised by porosity which may even reach 90%, one of the mechanisms of the heat transfer that affects its heating is the natural convection of air inside individual sections. Therefore, an optical visualisation of this phenomenon is very useful for its analysis. The Schlieren apparatus used for the examinations is composed of two parts which have special optical parts (lenses and mirrors). One of these parts generates a parallel light beam that interacts with the medium examined. Next, the light reaches the second part, equipped with optical devices that allow for obtaining the image of the phenomenon studied, which is displayed on the screen. A very important component of the apparatus is "an optical knife". The light that reproduces the image of the phenomenon is focused on the edge of the optical knife. Moving the knife edge allows for regulation of the contrast and brightness of the image which is displayed on the screen. A more detailed description of this apparatus was presented in [18].

The second component of this test stand is an electrical furnace located in the research space of the Schlieren apparatus. The construction of the furnace allows for one-dimensional heating of the samples while, at the same time, ensuring the unlimited passing of light through them. The general view of the stand is presented in Figure 10.

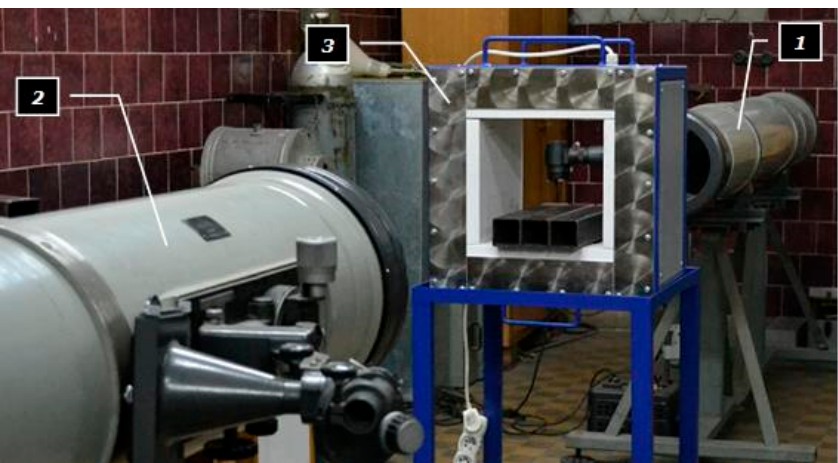

**Figure 10.** A stand for visualisation of convection phenomena using the Schlieren method: (1) part of the apparatus that generates the light beam; (2) part that displays the disturbed light beam, (3) electric furnace with the sample.

The results of the examinations performed for four samples made of various types of steel sections (Figure 11) are presented below. The sections were located in the furnace chamber directly over the resistant heating plate, parallel to the optical axis of the Schlieren apparatus. This allowed for the light to pass through the internal spaces of the sections. After starting the heater, the heat transfer in the samples was oriented vertically upwards. One of the mechanisms of the heat exchange was natural convection of air inside the sections. The main part of the examinations was an observation of the air movements during the heating. The images recorded for individual samples are presented in Figures 12–15.

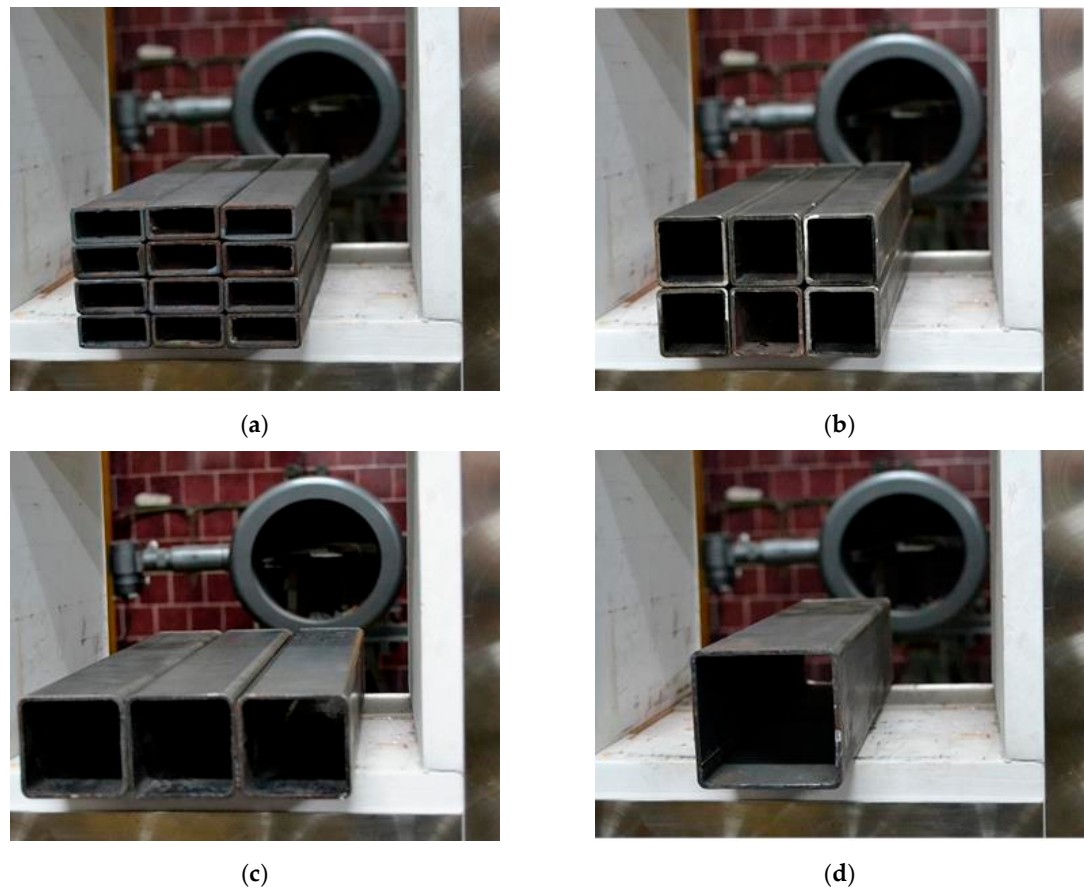

**Figure 11.** The samples examined by means of Schlieren imaging: (**a**) a bed of 20 × 40 mm sections; (**b**) a bed of 40 × 40 mm sections; (**c**) a layer of 60 × 60 mm sections; (**d**) a single 80 × 80 mm section.

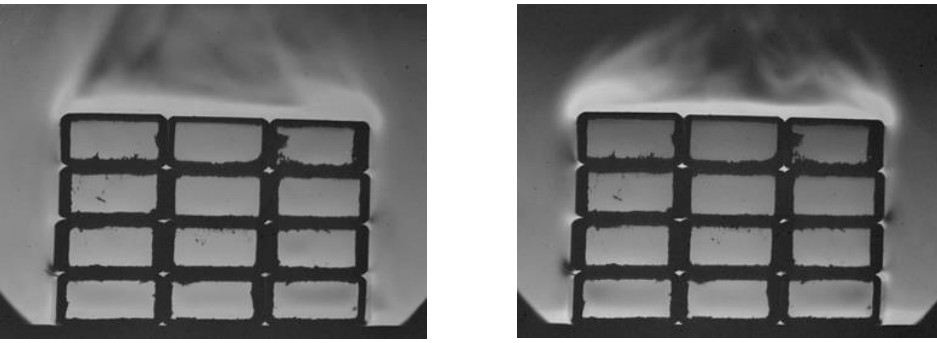

**Figure 12.** Schlieren images recorded during heating of the 20 × 40 mm sections.

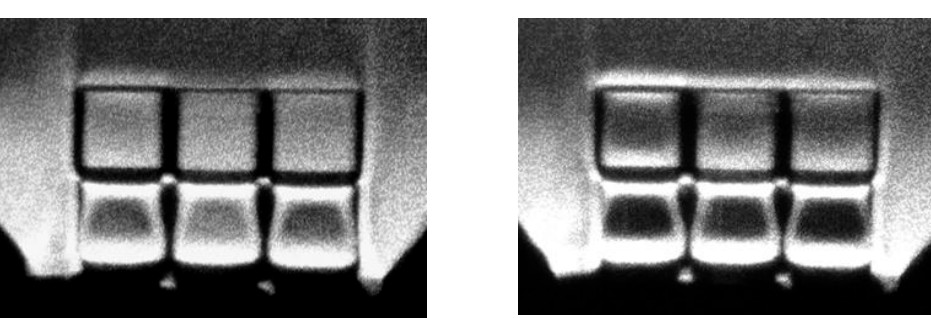

**Figure 13.** Schlieren images recorded during heating of the 40 × 40 mm sections.

Figure 12 illustrates the images obtained during the heating of the sample made of 20 × 40 mm sections. In this case, no air movements can be observed inside the sections. Contrary to the air outside the sample (which is substantially disturbed), the image is entirely homogeneous. A lack of air movement inside the sections results from their insignificant dimensions since it is entirely suppressed by boundary layers.

Figure 13 illustrates the images recorded for the 40 × 40 mm sections. Boundary layers are formed especially along the vertical walls, with their thickness increasing with time. However, apart from these layers, air remains still in the "nucleus". Therefore, natural convection inside these sections also does not occur.

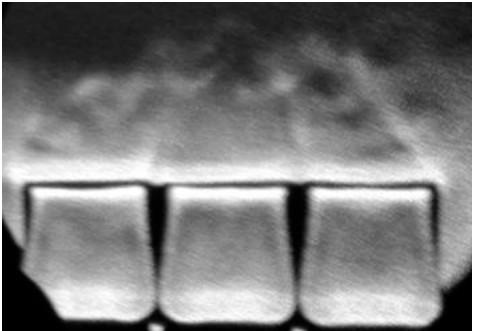 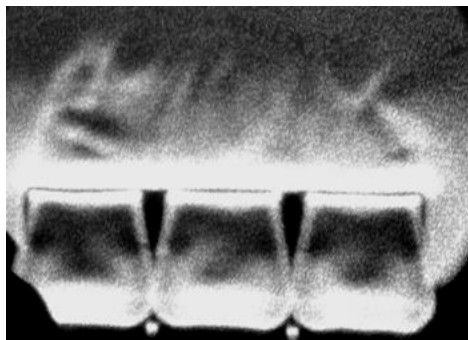

**Figure 14.** Schlieren images recorded during heating of the 60 × 60 mm sections.

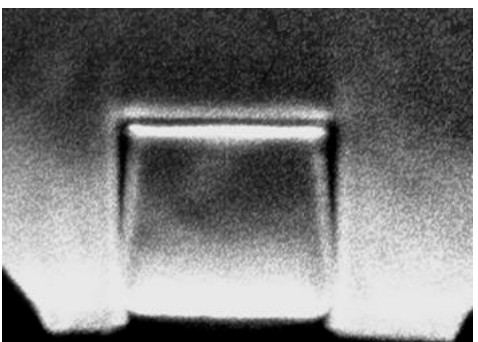 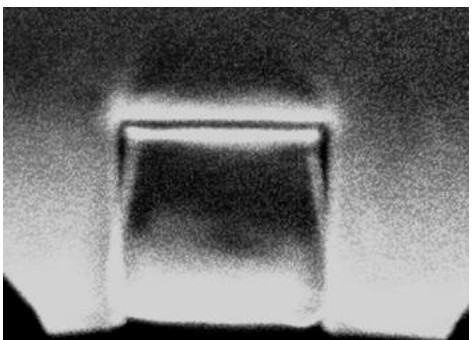

**Figure 15.** Schlieren images recorded during heating of the 80 × 80 mm section.

A different pattern was observed during the heating of the 60 × 60 mm sections, with images presented in Figure 14. Due to the greater dimensions of these elements, the air flow was observed after the formation of the boundary layers in the areas outside these layers. In this case, boundary layers are at a distance that prevents them from suppression of the convection movements. As shown in Figure 15, the same effect was observed for the heating of the 80 × 80 mm sections.

The study demonstrated that the intensity of natural convection inside rectangular sections mainly depends on the dimensions of these elements. Although the results of these analyses are very useful, they are only of a qualitative character. However, after supporting these findings with further research based on the dimensional analysis and the theory of similarity, it is possible to describe the natural convection in quantitative terms. For this purpose, the Rayleigh and Nusselt numbers are used [44,45].

## 4. The Investigation of a Porous Charge by Means of Thermography

Experimental examinations of the heat transfer in the porous charge with the use of thermography provide both qualitative and quantitative insight into this process. Qualitative information concerns the contribution of individual mechanisms of the heat transfer (conduction, contact conduction, free convection and radiation) that occur in the area of the charge. The real effect of these mechanisms is the temperature field, which is formed within the charge. Thus, based on the information about the temperature field in the charge

determined from measurements, it is possible to determine the intensity and contribution of particular mechanisms of the heat transfer. Furthermore, quantitative information is connected with opportunities to determine the effective thermal conductivity $k_{ef}$. This approach consists of calculating the $k_{ef}$ coefficient based on the information about temporal changes in the temperature field using the analysis of inverse heat conduction problems [46].

One example of using thermography to examine the porous charge is measuring the bed of 40 mm round bars, which was heated in the electrical chamber furnace. The charge was placed so that its face coincided with the plane of the furnace door. Thus, when the furnace was closed, the surface of the charge was in contact with the insulation of the internal surface of the furnace door. Therefore, heating this surface through convection was limited. In order to eliminate the radiation from the furnace interior that would disturb the measurements, any empty spaces were filled with the ceramic fabric. The view of this charge is presented in Figure 16a. After starting the furnace, thermograms were recorded at five-minute intervals, as presented in Figure 16b. The measurements were performed with the furnace door open, whereas after recording the thermogram, the door was closed. The heating time was 180 min. The temperature of the charge surface at the end of the process was over 500 °C. Two chosen thermograms recorded during these examinations are presented in Figure 17.

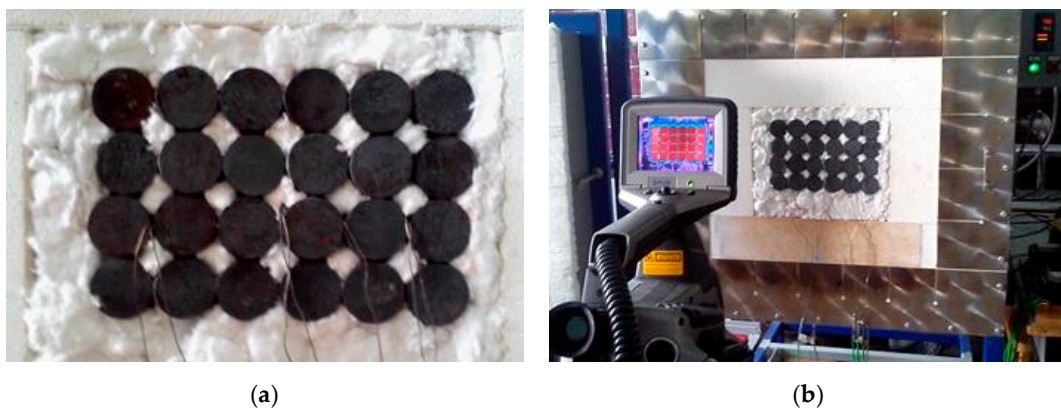

(**a**)          (**b**)

**Figure 16.** A bed of 40 mm bars in the electric furnace prepared for thermographic examinations: (**a**) view of the sample; (**b**) thermal camera recording the temperature of the heated charge.

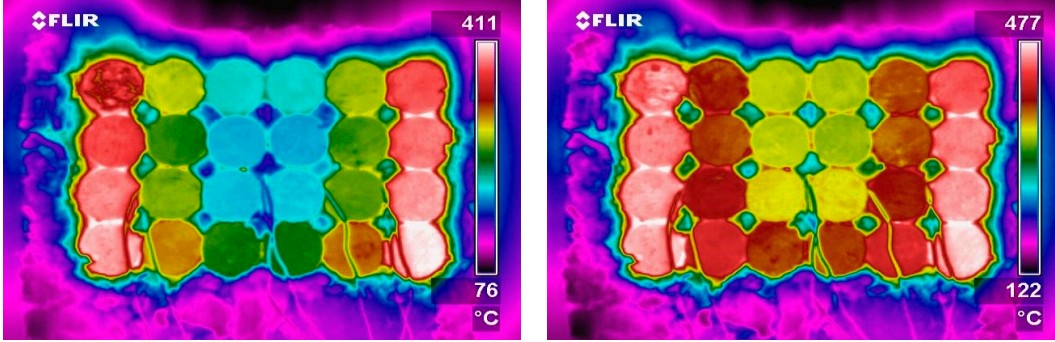

**Figure 17.** Selected thermograms recorded during heating of the charge presented in Figure 16.

The obtained thermograms were then analysed in detail. The analysis consisted of separating three areas (denoted as Ar1, Ar2 and Ar3) which were coincident with the surfaces of three bars from the third layer of the bed. An example of the thermogram with these areas is presented in Figure 18. Next, minimum temperature $t_{min}$, maximum temperature $t_{max}$, average temperature $t_{av}$ and the difference $\Delta t = t_{max} - t_{min}$ were determined.

As demonstrated, the thermographic method allows for a precise determination of the changes in the temperature field over the whole surface of the charge. In practice, there

is no other technique of temperature measurement that offers such opportunities. The data obtained through the analysis provide important insights into the analysed process. The bases for such analyses are the diagrams that present changes in the parameters determined versus time. Two such charts are presented in Figure 19. Figure 19a shows changes in $t_{av}$, whereas Figure 19b illustrates changes in temperatures variation $\Delta t$. These data allow determining the intensity of individual processes of the heat exchange (heat radiation, natural convection or contact conduction) that occur in the area of the charge. It is possible to achieve it by introducing the notion of thermal resistance for each type of heat transfer [47].

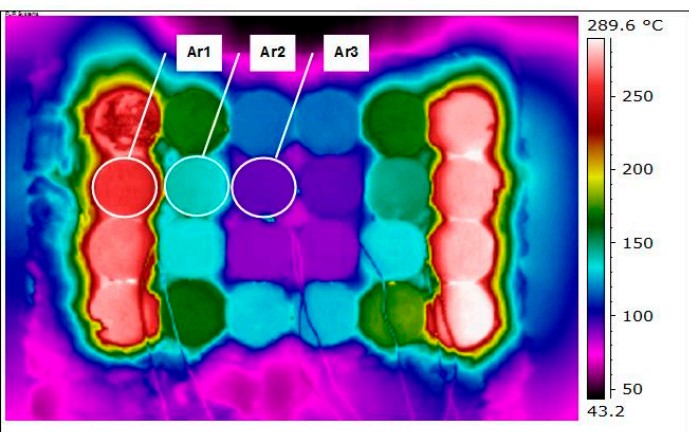

**Figure 18.** Thermogram of the charge from Figure 16 with marked areas Ar1–Ar3.

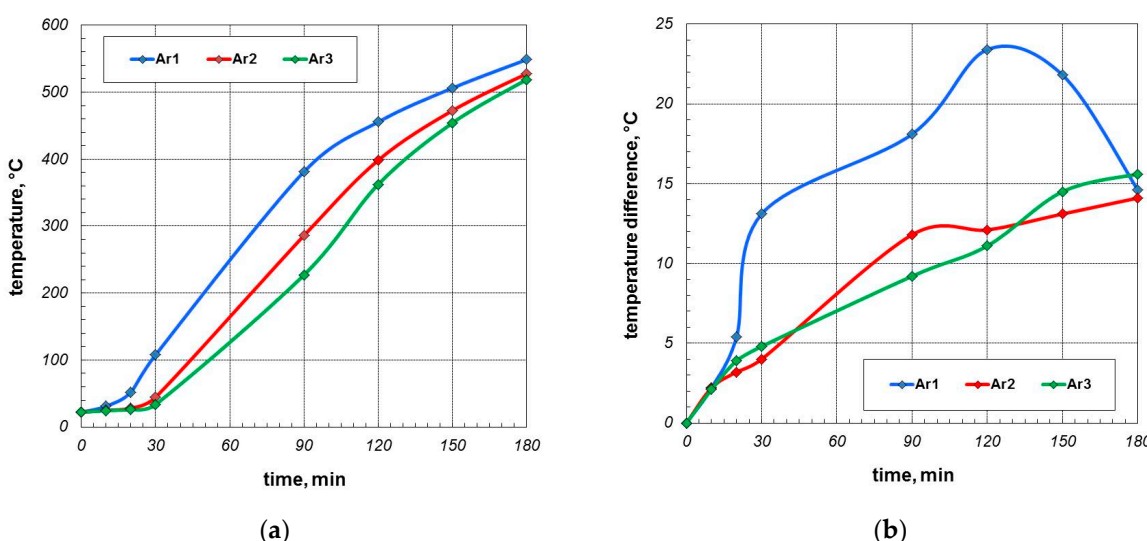

**Figure 19.** Changes of selected parameters determined for areas Ar1–Ar3: (**a**) changes in average temperature; (**b**) changes in the temperature difference $\Delta t$.

The second example of using thermography for examinations of heating the porous charge are measurements performed for the flat bed of 20 mm square bars. The fragment of the sample of such a bed is presented in Figure 20a. The bed was heated in the furnace, which ensured one-dimensional heat flow. During this process, a thermogram for the frontal surface was recorded from time to time. One of the thermograms is presented in Figure 20b. This thermogram contains three vertical lines Li1–Li3. Next, using the specialised software, changes in the temperature along these lines were evaluated, as illustrated in Figure 21a. The shape of individual lines in the chart is very irregular. Obviously, the temperature of a steel bar, which is a very good heat conductor, cannot be reflected by such a pattern. This effect is likely to result from differences in the emissivity of the bars caused by the oxide

layer present on their surfaces. In order to make further analysis easier, the results from Diagram 21a were smoothed, obtaining consequently the patterns shown in Figure 21b. The temperature declines are very easy to be quantitatively interpreted.

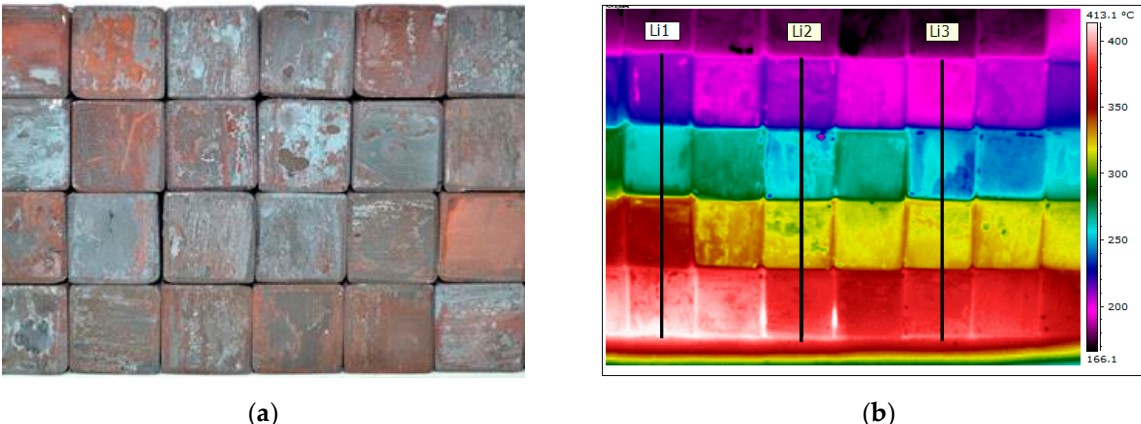

(**a**)          (**b**)

**Figure 20.** A bed of 20 mm square bars heated in the electric furnace: (**a**) view of the sample; (**b**) one of the thermogram recorded during the test.

As the results in Figure 20b show, the temperature decline in the area of the bed was determined based on the height of four layers. Therefore, temperature changes presented in Diagram 13b concern the declines that occur alternately in the area of layers of bars and in joints. As can be observed, greater declines concern joints. This seems obvious since joints are locations of the highest heat flow resistance, with its measure provided by the temperature decline.

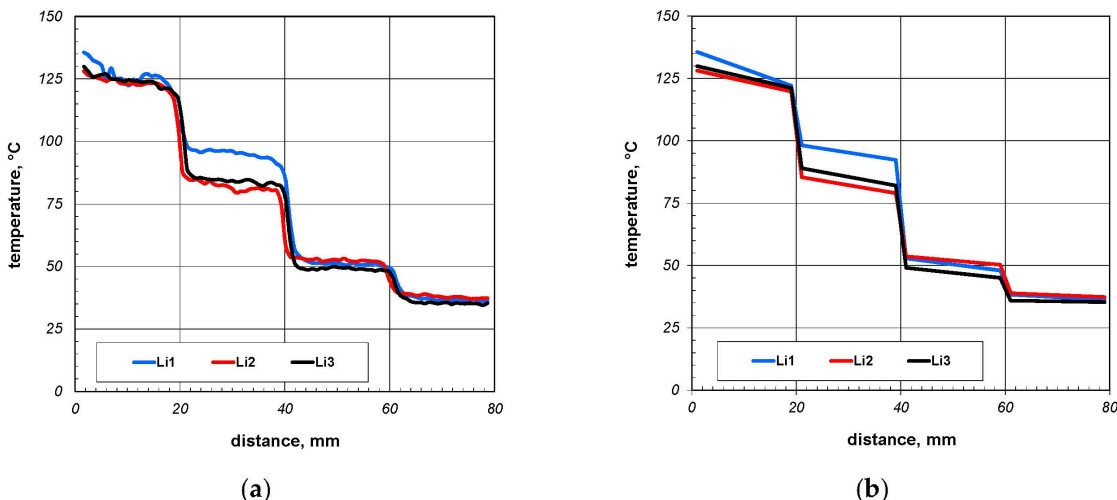

(**a**)          (**b**)

**Figure 21.** (**a**) Changes in temperature along the line Li1–Li3 from the thermogram 20b; (**b**) temperature changes along the Li1–Li3 line after smoothing.

If the thermogram is recorded under conditions of steady heat transfer, it can be used for the evaluation of several thermal parameters. The resistance for heat conduction occurring in the area of bars $R_{br}$ can be expressed by means of the relationship that results from the Fourier equation (Equation (6)). Furthermore, if the thermal conductivity of bars $k_{br}$ is known, Equation (6) can be used to determine the heat flux (Equation (7)). If the value of the heat flux is known, the thermal resistance in the bars joints $R_j$ can be determined (Equation (8)):

$$R_{br} = \frac{\Delta t}{q} = \frac{l_{br}}{k_{br}}, \tag{6}$$

$$q = k_{br}\frac{\Delta t}{l_{br}} \tag{7}$$

$$R_j = \frac{\Delta t_j}{q} \tag{8}$$

where: $\Delta t$—difference in temperature in the bar cross-section; $q$—heat flux in the sample; $l_{br}$—dimensions of bars in the direction of heat flow; $k_{br}$—thermal conductivity of bars; $\Delta t_j$—temperature drop in the area of the joint.

The investigations according to the presented methodology for a bed of square 20 mm bars were described in [48]. It was established that $R_j$ resistance for the tested sample changed from 0.00149 to 0.00280 (m²·K)/W. An increase in temperature of the sample decreased the value of $R_j$, which is determined by the phenomenon of the heat radiation.

The resistances $R_{br}$ and $R_j$ can be used to determine the value of the effective thermal conductivity $k_{ef}$ of the charge. This is achieved through the analysis of series and parallel connections of heat resistances for a specifically separated small part of the medium analysed, which is called a unit cell. This methodology is widely used in the theory of porous media for the determination of effective thermal conductivity [49–51].

## 5. Modelling of Effective Thermal Conductivity of a Porous Charge

The literature presents numerous analytical models of the effective thermal conductivity dedicated to solid body-gas [23–25]. They can be divided into two general groups: simple models and complex models [52]. Simple models predict the value of the $k_{ef}$ coefficient only as a function of the primary parameters, i.e., thermal conductivities of the solid phase $k_s$ and gas phase $k_g$ and the porosity of the medium $\varphi$, whereas complex models include additional parameters, also called secondary parameters. This group of parameters includes: contact resistance, thermal radiation as well as the mean size of grains or voids. The usefulness of the chosen models of effective thermal conductivity for determining the thermal properties of a porous charge was analysed in [53]. The paper analyses nine simple models (Series, Parallel, Maxwell–Eucken, EMT, Horai, Beck, Assad, Woodside and Bruggeman models) and two complex models (Kunii–Smith and Zehner–Bauer–Schlünder models). It was stated that simple models do not provide correct results, which are several times greater or smaller than the experimental $k_{ef}$ values. Complex models proved to be much more useful. However, obtaining the correct $k_{ef}$ values with the use thereof requires a careful fit of the parameters, which expresses the intensity of thermal contact conduction. However, obtaining a correct fit requires measurement data, which is a major disadvantage of these models. This leads to a general conclusion that the crucial problem in modelling the effective thermal conductivity of a steel porous charge with the use of complex models is the proper description of the thermal contact conduction.

Apart from the above-mentioned models, which have a general application, the literature provides also models dedicated to a steel porous charge [54]. Given equations concern the three types of a charge with a layered (Equation (9)), fibrous (Equation (10)) and granular structure (Equation (11)) with the following forms:

$$k_{ef} = \frac{k_{eq}}{1 - (1 - k_{eq})(1 - \varphi)}k_s, \tag{9}$$

$$k_{ef} = \frac{1 - \varphi(1 - k_{eq})}{1 - \varphi(1 - k_{eq})(1 - \varphi)}k_s \tag{10}$$

$$k_{ef} = \frac{1 - \varphi^2(1 - k_{eq})}{1 - \varphi^2(1 - k_{eq})(1 - \varphi)}k_s \tag{11}$$

$$k_{eq} = \frac{k_g + h_{rd}\delta}{k_s}, \tag{12}$$

$$h_{rd} = 0.186\left(\frac{T_m^3}{100}\right) \tag{13}$$

where: $k_s$, $k_g$—thermal conductivities of steel and gas, respectively; $\delta$—thickness of the metal layer; $T_m$—absolute mean temperature of the charge.

Using Equations (9)–(13), the computations of the $k_{ef}$ coefficient were made for the temperatures of 25–700 °C. The analysis dealt with four porosities (0.1, 0.21, 0.35 and 0.5), two values of metal layer thickness (10 and 40 mm), for which the thermal conductivities for steel $k_{st}$ and gas $k_g$ change according to the following equations [47]:

$$k_s = 1.24\cdot10^{-8}t^3 - 3.26\cdot10^{-5}t^2 - 1.19\cdot10^{-2}t + 51.35, \tag{14}$$

$$k_g = -2.88\cdot10^{-8}t^2 + 8.05\cdot10^{-5}t + 0.02 \tag{15}$$

The values of the $k_{ef}$ coefficient presented in Figure 22 concern the charge with the layered structure. Figure 22a presents the results for the layers of metal with the thickness of 10 mm, whereas Figure 22b for the thickness of 40 mm. The effective thermal conductivity calculated for this model increases with temperature and decreases with the charge porosity. In the case of a 10 mm layer, $k_{ef}$ ranges from 0.15 to 11.8 W/(m·K), and in the case of a 40 mm layer, the range is 0.44–23.1 W/(m·K).

The model values of $k_{ef}$ obtained for the charge with a fibrous structure are presented in Figure 23. In this case, the effective conductivity decreases with temperature and insignificantly depends on the $\delta$ parameter (results in Figure 23a,b are nearly identical). The effective thermal conductivity, in this case depending on temperature and porosity, ranges from 21.6 to 50.4 W/(m·K). Very similar results were obtained for the granular structure model, presented in Figure 24. This model, compared to the previous one, gives very similar results; however, it is only slightly less sensitive to changes in porosity. The effective thermal conductivity obtained in this case ranges from 27.1 to 50.9 W/(m·K).

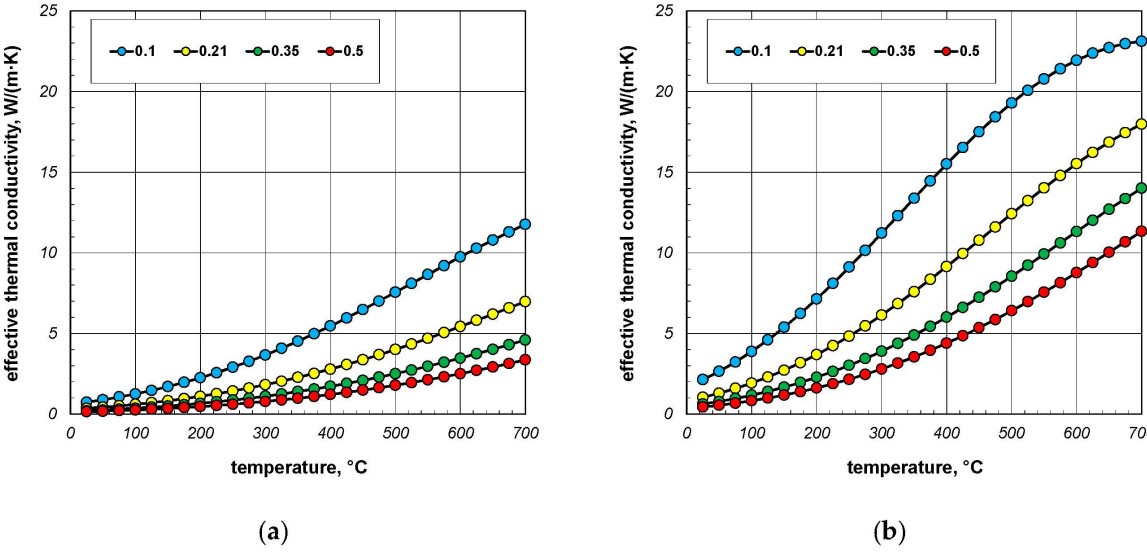

**(a)**            **(b)**

**Figure 22.** Calculated values of the effective thermal conductivity for a porous charge of a layered structure: (**a**) results for $\delta$ = 10 mm; (**b**) results for $\delta$ = 40 mm. The values visible in the key refer to the porosity of the charge.

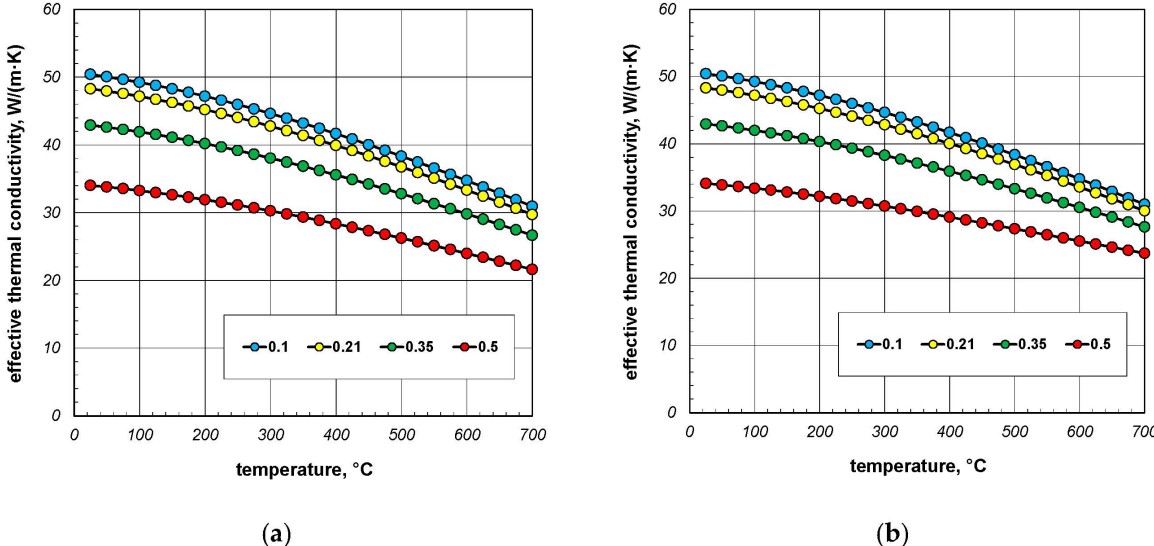

(**a**)                                    (**b**)

**Figure 23.** Calculated values of the effective thermal conductivity for a porous charge of a fibrous structure: (**a**) results for $\delta$ = 10 mm; (**b**) results for $\delta$ = 40 mm. The values visible in the key refer to the porosity of the charge.

The values of the $k_{ef}$ coefficient obtained from the three analysed models are not consistent with the presented earlier experimental results. The models concerning a fibrous and granular charge do not even yield values correct from a qualitative point of view. The $k_{ef}$ coefficient obtained from these models decreases with temperature, whereas it should actually increase. This tendency was observed only for the model of the layered charge. However, it is not very accurate in quantitative terms. As demonstrated by the measurements for the charges analysed (with the porosity of 0.1–0.88), the effective thermal conductivity ranges from 1.05 to 6.9 W/(m·K), whereas for the model (with the porosity of 0.1–0.5), this range is from 0.15 to 23.1 W/(m·K). Therefore, this model also seems to be useless from the practical point of view.

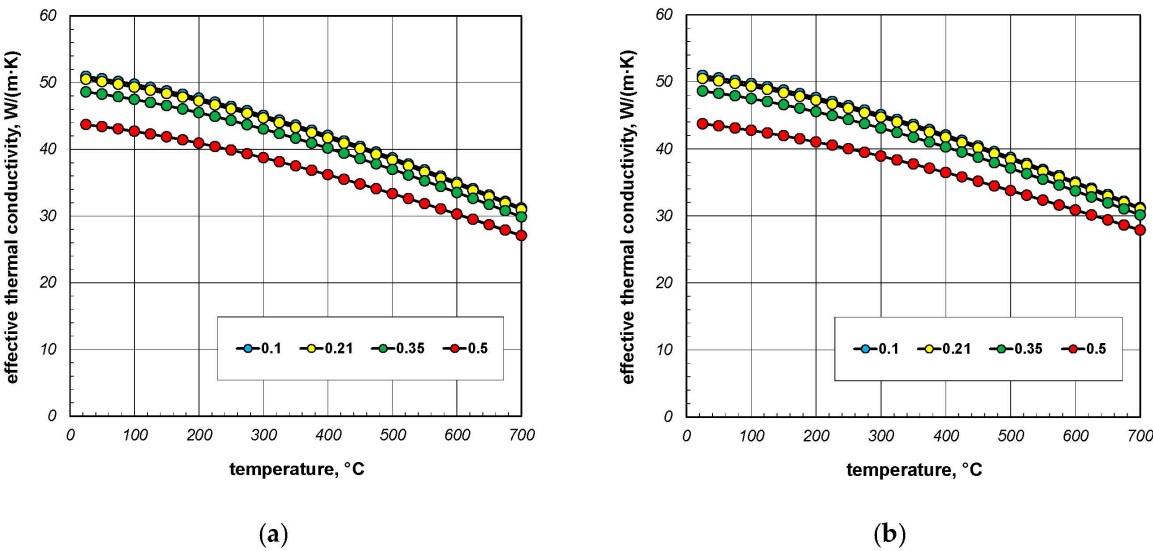

(**a**)                                    (**b**)

**Figure 24.** Calculated values of the effective thermal conductivity for a porous charge of a granular structure: (**a**) results for $\delta$ = 10 mm; (**b**) results for $\delta$ = 40 mm. The values visible in the key refer to the porosity of the charge.

The presented results of model calculations of the $k_{ef}$ coefficient lead to a conclusion that the literature models are not suitable for determining the thermal properties of a

porous charge. In order to do that, it is necessary to apply individually developed models, which take into account the characteristics of the heat transfer which occurs in the analysed medium. A very convenient approach in this situation is the thermal resistance concept or the electrical analogy. This method can be used to solve steady heat transfer problems that involve the analysis of combined series-parallel arrangements. The starting point for such an analysis is a geometrical model of the analysed medium in the form of a so-called elementary cell. Such a cell is made up of a repeatable fragment of the considered charge. Then, the total thermal resistance $R_{tot}$ for such a cell is obtained. Finally, the effective thermal conductivity is calculated based on the equation for the conduction resistance of the plane wall [38]:

$$k_{ef} = \frac{L}{R_{tot}}, \tag{16}$$

where $L$ is the dimension of the cell in the heat flow direction.

A model describing the total thermal resistance for a complex heat transfer in the package of steel rectangular sections which uses the electrical analogy was described in the article [55]. When determining the thermal resistance $R_{tot}$, this model takes into consideration the following types of heat transfer: conduction in section walls, conduction and natural convection within gas, heat radiation between the walls of a section, as well as contact conduction between the adjacent sections. The total thermal resistance of the system considered in this case is considered as a serial connection of two resistances: heat resistance of sections $R_{st}$ and thermal contact resistance $R_{ct}$ that occurs between the adjacent layers of the package:

$$R_{tot} = R_{st} + R_{ct}, \tag{17}$$

Determining the resistance $R_{st}$ requires taking into consideration all the mechanisms of the heat flow that take place in the area of the sections. The heat in this element is transferred by: conduction in steel walls, conduction and free convection within gas inside the sections and thermal radiation between the inner surfaces of a section. Each of the individual heat transfer mechanisms is assigned a corresponding thermal resistance. For this reason, when calculating the value of $R_{st}$, it is necessary to take into consideration the following: conduction resistances in steel $R_s$, conduction resistance in gas $R_{gs}$ and radiation resistance $R_{rd}$. This solution assumes a one-dimensional heat transfer characterised by a heat flux $q$. In this situation, a complex multidimensional heat transfer is regarded as one-dimensional. Due to such a simplification, the two following assumptions are used: (i) any plane wall normal to the direction of the heat flow is isothermal; (ii) any plane parallel to the direction of the heat flow is adiabatic [38]. This approach leads to two different resistance networks, which also means two different values for the total thermal resistance $R_{st-a}$ and $R_{st-b}$.

The resistance $R_{st-a}$ relates to the division of the section into three vertical zones *I-III* parallel to the direction of heat flow (Figure 25a). The resistance $R_{st-b}$ relates to the division of the section into three horizontal layers 1–3 (Figure 25b). The thermal resistance networks for both divisions of the section are shown in Figure 24. On this basis, it is possible to write down:

$$R_{st-a} = \left( \frac{1}{R_I} + \frac{1}{R_{II-1} + \left( \frac{1}{R_{gs}} + \frac{1}{R_{rd}} \right)^{-1} + R_{II-3}} + \frac{1}{R_{III}} \right)^{-1}, \tag{18}$$

$$R_{st-b} = R_1 + \left( \frac{1}{R_{2-I}} + \frac{1}{R_{gs}} + \frac{1}{R_{rd}} + \frac{1}{R_{2-III}} \right)^{-1} + R_3 \tag{19}$$

As each division of a section takes into account different assumptions about the temperature field, the value of $R_{st-a}$ is slightly bigger than the value of $R_{st-b}$. For this reason, the final value of $R_{st}$ is calculated as follows [56]:

$$R_{st} = \frac{R_{st-a} + 2R_{st-b}}{3}, \tag{20}$$

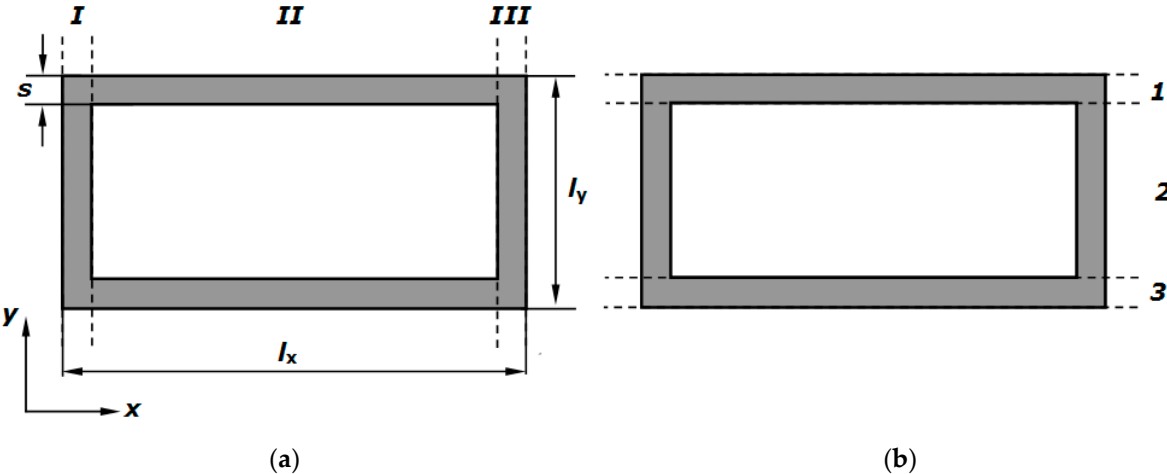

**Figure 25.** Division of a section into: (**a**) vertical zones *I-III*; (**b**) horizontal layers 1–3.

The resistances occurring in Equations (18) and (19), which apply to conduction in section walls, are marked grey in Figure 26. The resistance $R_{gs}$ applies to the heat transfer in the gas. This phenomenon is connected with simultaneous conduction and natural convection. Both mechanisms of the heat transfer can be treated jointly as an intensified heat conduction expressed quantitatively by the equivalent gas thermal conductivity $k_{eg}$ [38,45]:

$$k_{eg} = k_g Nu, \tag{21}$$

where: $k_g$—thermal conductivity of gas; $Nu$—Nusselt number. The methodology of determining the $Nu$ number for natural convection within a steel section was described in [45]. Depending on the section size and gas temperature, the value of $Nu$ varies within the range from 1.2 to 7.1.

Radiation resistance $R_{rd}$ is obtained on the basis of the analysis of thermal radiation exchange in the system which consists of four flat surfaces that close a space. The methodology for the determination of $R_{rd}$ resistance for this system was described in [57].

The thermal contact resistance $R_{ct}$ from Equation (17), with respect to temperature $t$, can be approximated by the polynomial [55]:

$$R_{ct} = \left( A_1 t^2 + A_2 t + A_3 \right) \cdot 10^{-4}, \tag{22}$$

Based on the experimental investigations, it has been established that due to shape errors of the sections, the value of $R_{ct}$ resistance can fit within certain limits. The minimum and maximum values of the resistance $R_{ct}$ are described by the following equations:

$$R_{ct-\min} = \left( 1.25 \cdot 10^{-5} t^2 - 0.0288t + 32.91 \right) \cdot 10^{-4}, \tag{23}$$

$$R_{ct-\max} = \left( 2.31 \cdot 10^{-5} t^2 - 0.0534t + 58.16 \right) \cdot 10^{-4} \tag{24}$$

Using Equations (23) and (24) for the considered section package, two limiting values of the effective thermal conductivity $k_{ef}$ can be received. The $k_{ef}$ coefficient of a given section package should remain between these values. Such an approach is possible because this coefficient is not a material property but expresses only the ability of a given charge to transfer heat. In the case of the analysed charge, this feature greatly depends on the contact conditions in adjacent sections' layers, which may vary for different section types.

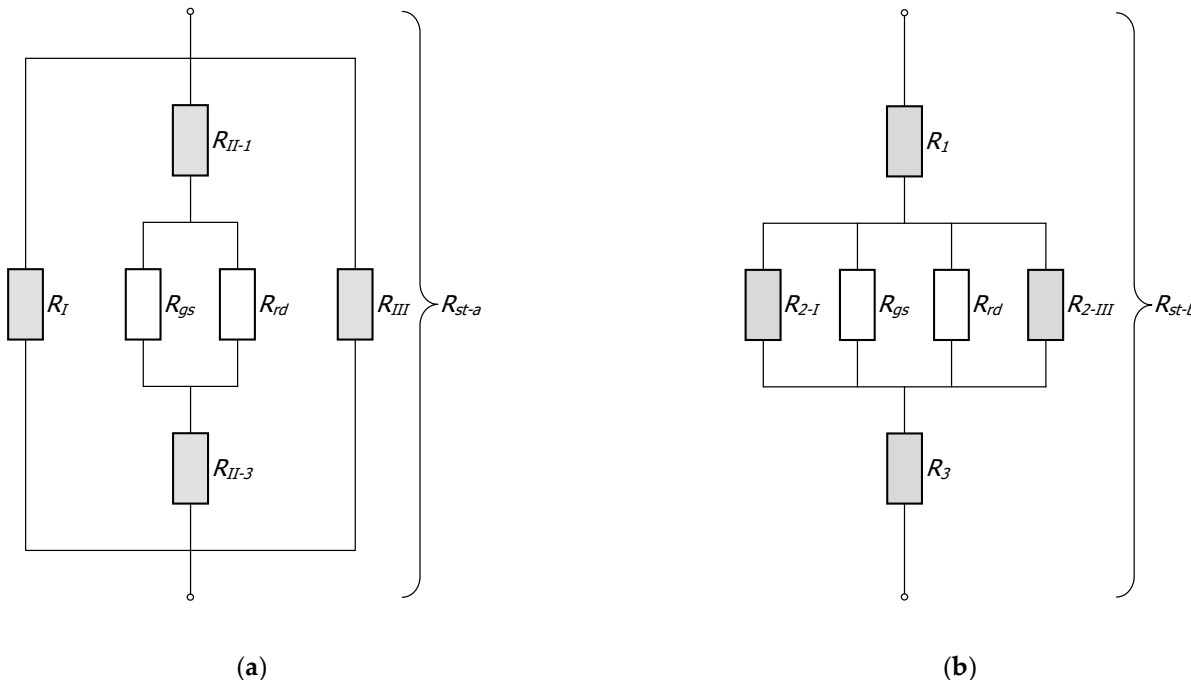

**Figure 26.** The networks of thermal resistance used for a rectangular section: (**a**) network for $R_{st-a}$ resistance; (**b**) network for $R_{st-b}$ resistance.

The results of calculations of the $k_{ef}$ coefficient according to the presented methodology obtained for the two different packages of profiles are presented in Figure 27. Figure 27a shows the results obtained for a package of $20 \times 40$ mm sections with a wall thickness of 2.5 mm. The results in this case refer to a situation when a heat flow occurred along a shorter wall of the section. The above-mentioned diagram additionally shows the results of experimental measurements of the $k_{ef}$ coefficient of this charge. As can be seen, the maximum values of $k_{ef}$ obtained for the minimum value of the $R_{ct}$ resistance (Equation (23)) are between the experimental values. A similar situation was observed, in the case, of a bundle of $60 \times 60$ mm sections, for which the results of model calculations and measurement of the $k_{ef}$ coefficient are presented in Figure 27b. In this case, the wall thickness was adopted as 3 mm.

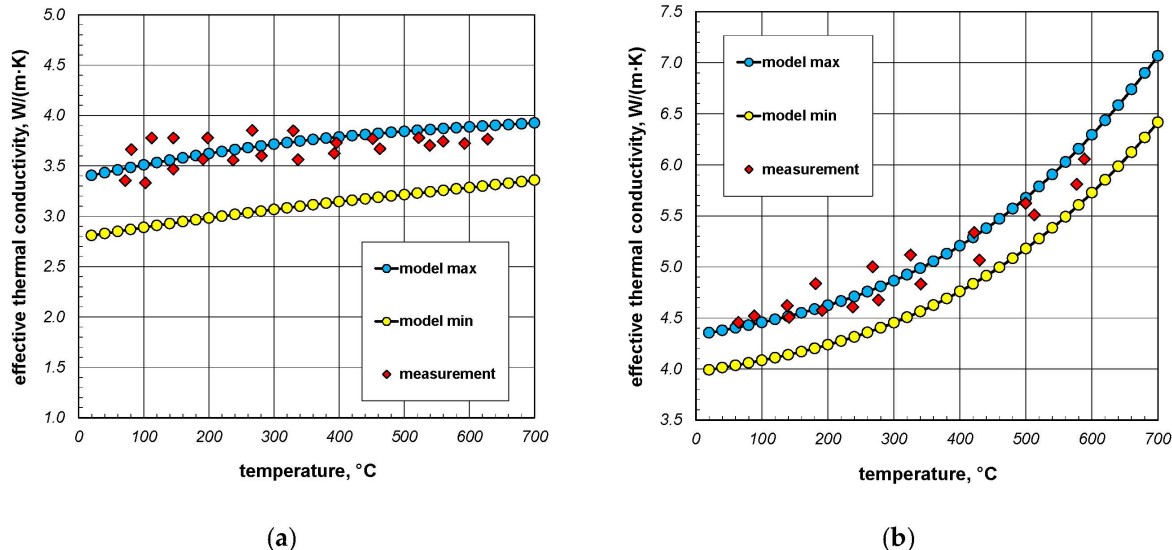

**Figure 27.** A comparison of model and measurement values of the effective thermal conductivity of section packages: (**a**) $20 \times 40$ mm sections; (**b**) $60 \times 60$ mm sections.

In order to assess the quality of the results obtained with the use of the presented model, it was necessary to calculate the percentage discrepancies between the measurement and model values of the $k_{ef}$ coefficient. This parameter was defined with the use of the following relationship:

$$dk_{ef} = \frac{\left| k_{ef-ex}(t_i) - k_{ef-\text{mod}}(t_i) \right|}{k_{ef-ex}(t_i)} \cdot 100\%, \tag{25}$$

Table 2 shows the minimum, mean and maximum values of $dk_{ef}$ obtained for the analysed section packages. The obtained values of $dk_{ef}$ show that the differences between the model and experimental results for both packages fit within the range of measurement uncertainty of the $k_{ef}$ coefficient. Therefore, it can be stated that the proposed model of effective thermal conductivity provides results that are very close to the values obtained during measurements. What is important, this model also correctly predicts the changes of the $k_{ef}$ coefficient in the temperature function. This example proves that the method of analysis of the complex heat transfer in a porous charge based on electrical analogy is an effective computational tool.

**Table 2.** The minimum, mean and maximum values of $dk_{ef}$ defined by Equation (25).

| Type of Sections in the Package | $dk_{ef}$, % | | |
| --- | --- | --- | --- |
| | Minimum | Mean | Maximum |
| 20 × 40 mm | 1.21 | 3.69 | 7.17 |
| 60 × 60 mm | 0.02 | 2.78 | 6.11 |

## 6. Conclusions

Investigations described in the previous points aim at understanding the qualitative and quantitative mechanisms of heat transfers that occur during the heat treatment of porous charge. Based on the described investigations, it was possible to determine the values of the following parameters: effective thermal conductivity, thermal contact resistance and Nusselt number. The obtained knowledge enables developing a model of the effective thermal conductivity for the given charge. Using the model of thermal conductivity, it is possible to analyse the transient temperature field in the treated charge. This type of model, based on the method of energy balance dealing with a charge of rectangular steel bars, was described in [58]. It is a convenient tool, the results of which can contribute to the reduction in energy consumption and emission of pollutants as well as to the increase in production capacity and quality improvement of the finished products. Therefore, investigations of the heat transfer in steel porous charge are of major importance for the industry. In future papers, the authors will investigate thermal contact conductance as it is the main mechanism that influences the course of thermal treatment of a porous charge.

**Author Contributions:** Conceptualisation, R.W. and M.G.; methodology, R.W. and V.B.; measurements, R.W.; validation, M.G. and R.W.; formal analysis, R.W., V.B. and P.A.K.; writing—original draft preparation, R.W., V.B. and M.G.; writing—review and editing, R.W., V.B. and P.A.K.; visualisation, R.W. and V.B.; supervision, R.W. All authors have read and agreed to the published version of the manuscript.

**Funding:** This research received no external funding.

**Institutional Review Board Statement:** Not applicable.

**Informed Consent Statement:** Not applicable.

**Data Availability Statement:** Not applicable.

**Conflicts of Interest:** The authors declare no conflict of interest.

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
