# Peer review of "The Review of Chosen Methods Used to Investigate Heat Transfer in a Steel Porous Charge"

_energies, doi:10.3390/en15062266_

Round 1

Reviewer 1 Report

Article Reviewed:

Journal: Energies (MDPI)

Title: The review of chosen methods used to investigate heat transfer in a steel porous charge Manuscript ID: Energies-1634524

Decision: Minor Revision

Summary:

It is interesting article and have many applications in the polymers science and technology. The presentation of the article is very good and it falls under the aims and scope of the journal.

Recommendation:

The manuscript is well written and presented. The paper is strongly recommended for publication in the journal Energies.

Some observations, although not strictly mandatory, might however help in further improving the quality of the submission.

I can strongly recommend it for publication in Energies with the following minor revision.

  1. Abstract should be included some key outcomes.
  2. Keywords should be specific, reduce them.
  3. Provide the nomenclature
  4. Check the article thoroughly for the grammatical errors.
  5. Literature survey looks poor on the current study. Include the following recent relevant references of the current study.
  • https://doi.org/10.3390/coatings11121552

Reviewer 2 Report

The study is related to heat transfer and is interesting. The main problem is clearly presented. The
manuscript investigates the main question and have described the results in conclusions. The
manuscript needs a minor revision for the recommendation to be publish in the Journal Energies.
(1) The first three sentences of the abstract must be removed. The abstract must be started with
“The paper presents...”.
(2) The fist sentence of the introduction part is wrong. Make correct this sentence. Also, the
introduction is weak. Read and cite the following relevant papers. https://dx.doi.org/10.1002/zamm.202000212
(3) If the porosity quantities are enhanced, then what will happen?.
(4) The authors should focus on Cattaneo-Christov heat and mass flux theory in their future work.
(5) Provide the comparison with published work.
(6) Add the Nomenclature.
Thanks

Reviewer 3 Report

I recommend accepting the manuscript after the authors do the major revisions included in the review report in the attachment file.

Round 2

Reviewer 3 Report

After checking through the revised version, it is worth mentioning that the authors have satisfactorily responded to all the questions and made the necessary changes to the manuscript. I have no further questions and suggest the acceptance of the revised manuscript.